# Bioaffinity Fishing Procedure Using Secretory Phospholipase A2 for Screening for Bioactive Components: Modulation of Pharmacological Effect Induced by sPLA2 from *Crotalus durissus terrificus* by Hispidulin from *Moquiniastrum floribundum*

**DOI:** 10.3390/molecules25020282

**Published:** 2020-01-09

**Authors:** Adeilso Bispo dos Santos Junior, Cinthia I. Tamayose, Marcelo J. P. Ferreira, Mariana N. Belchor, Caroline R. C. Costa, Marcos Antonio de Oliveira, Marcos Hikari Toyama

**Affiliations:** 1BIOMOLPEP group, Instituto de Biociências, UNESP, Campus do Litoral Paulista, São Vicente CEP 11380-972, São Paulo 11380-972, Brazil; grimoriojunior@gmail.com (A.B.d.S.J.); mary_novo@hotmail.com (M.N.B.); carolsbert@gmail.com (C.R.C.C.); 2Departamento de Botânica, Instituto de Biociências, Universidade de São Paulo, São Paulo CEP 05508-090, Brazil; cinthiatamay@gmail.com (C.I.T.); marcelopena@ib.usp.br (M.J.P.F.); 3LABIMES, Instituto de Biociências, UNESP, Campus do Litoral Paulista, São Vicente CEP 11380-972, São Paulo 11380-972, Brazil; scaffix@gmail.com

**Keywords:** ultrafiltration, anti-inflammatory, natural compounds, bioaffinity

## Abstract

Bioaffinity capturing of molecules allows the discovery of bioactive compounds and decreases the need for various stages in the natural compound isolation process. Despite the high selectivity of this technique, the screening and identification methodology depends on the presence of a protein to capture potential ligands. However, some proteins, such as snake secretory phospholipase A2 (sPLA2), have never been investigated using this approach. The purpose of this study was to evaluate the use of a new method for screening natural compounds using a bioaffinity-guided ultrafiltration method on *Crotalus durissus terrificus* sPLA2 followed by HPLC-MS to identify the compounds, and this method could be used to discover new anti-inflammatory compounds from the various organisms originating from biodiversity. Different extracts were selected to evaluate their ability to inhibit sPLA2 activity. The extracts were incubated with sPLA2 and the resulting mixture was ultrafiltrated to elute unbound components. The resulting compounds were identified by HPLC-MS. We identified hispidulin as one of the components present in the *Moquiniastrum floribundum* leaf and evaluated the ability of this isolated compound to neutralize the inflammatory activity of sPLA2 from *Crotalus durissus terrificus*.

## 1. Introduction

Secretory A2 phospholipases (sPLA2) from snake venom in species such as *Crotalus durissus terrificus* (Cdt) have structural and functional similarity to most human sPLA2, and both have common enzymatic functions and properties [1]. These sPLA2 enzymes have been used as research tools both in silico and experimentally in the search for new anti-inflammatory compounds. sPLA2 is a pro-inflammatory protein involved in the mobilization of arachidonic acid (AA) through the indirect activation of cytosolic phospholipase A2 (cPLA2) and phospholipase C (PLC). These enzymes also lead to cyclooxygenase 2 (COX-2) activation, which is responsible for AA metabolism, releasing several pro-inflammatory mediators including prostaglandins and interleukins. In addition to AA, another important product for sPLA2-induced pharmacological activities is lysophospholipids, which are involved in the inflammatory process [2,3]. Both mediators induce an increase in the cytosolic expression of COX-2 and the consequent increase in prostaglandin synthesis [4,5]. The excessive release of free fatty acids by the action of both cytosolic PLA2 as well as secretory sPLA2 increases the activity of COX-2, which can result in formation of more free radicals and pro-inflammatory cytokines, leading to increased oxidative stress and early onset of inflammation [6,7,8]. Thus, sPLA2 is able to induce inflammation through its ability to generate AA, initiating a series of cellular events and inducing an increase in reactive oxygen species (ROS).

Phenolic compounds are considered one of the groups of bioactive compounds that have beneficial effects on human health. Phenolic compounds are known for their antioxidant activity due to a molecular structure capable of neutralizing or scavenging ROS and chelating metal ions. Thus, searching for natural compounds is important for identifying compounds with therapeutic potential, whether for the development of new antibiotics or new anti-inflammatory drugs [8,9].

Asteraceae is a family of eudicots characterized by diverse bioactive metabolites, and the *Moquiniastrum* genus, which belongs to this family, is found in South America and is used in folk medicine for treating respiratory diseases, such as asthma, bronchitis, coughs, and colds. Studies revealed these plants exhibit anti-trypanosomal, antiradical, anti-inflammatory, antispasmodic, antibacterial, cytotoxic, and antioxidant activities [10,11,12,13]. This genus has already been found to contain compounds such as genkwanin, hispidulin, cirsimaritin, and chlorogenic acid derivatives. Chlorogenic acids, present in coffee and green and black tea, in addition to *Moquiniastrum floribundum*, are associated with antioxidant and anti-inflammatory activity [13]. However, studies on *Moquiniastrum floribundum* compounds and their activity against sPLA2 anti-inflammatory activity are lacking.

Several strategies are used to screen for new secondary metabolites, including techniques that consider the interaction between natural compounds and ligands, which include biomolecules such as proteins, carbohydrate-bound peptide fragments, and other target compounds. The immobilization of these binders in a matrix or stationary phase enables the formation of the composite fishing system called ligand fishing [14,15,16]. In the offline fishing mode for natural compounds, the mixture of metabolites should be incubated with biomolecules, such as enzymes or receptors. After this interaction process, the bioactive compounds are retained by the binder and finally analyzed. In online mode, the extract containing the secondary metabolites is eluted through a chromatographic column containing a stationary phase immobilized with a binder [15,17]. Offline compound fishing and bioaffinity ultrafiltration have been widely used in the screening of active compounds from traditional Chinese medicines [18]. In this paper, we describe an alternative method for phytochemical screening and natural secondary metabolites using bioaffinity fishing using sPLA2 from Cdt.

## 2. Results

### 2.1. Evaluation of Plant Extracts on Edema Induced by sPLA2

Firstly, we investigated the anti-inflammatory effect of the different extracts obtained from *Moquiniastrum floribundum* (Cabrera) G. Sancho (Asteraceae) leaves, also called Cambará or vassourão branco. For these assays, we used a mass-to-mass concentration of extract and sPLA2 or commercial antivenom (Pró-Rural Produtos Agropecuário LTDA, Campo Grande, MS, Brazil) and sPLA2. Both antivenom and plant extracts were administered intraperitoneally to animals according to the standard method used for screening of anti-inflammatory compounds. Figure 1A shows the effect of injected antivenom 10 min after application of Cdt sPLA2, which significantly decreased the pro-inflammatory action induced by sPLA2. Figure 1B,D,E,G shows that hexane (Hex), ethyl acetate (AcOEt), *n*-butanolic (*n*-BuOH), hydromethanolic (HE), and methanol (MeOH) extracts significantly reduced sPLA2-induced edema. The results strongly suggest that these extracts should have a potential anti-inflammatory compound interacting with sPLA2. Dichloromethane extract (DCM; Figure 1C) produced no significant effect on sPLA2 edematogenic activity, whereas MeOH (Figure 1F) only produced a significant effect at the peak of edema.

### 2.2. Enzymatic Activity of sPLA2 in the Presence of Extracts

Figure 2 depicts the relationship of the various plant extracts with the enzymatic activity of sPLA2 from Cdt. According to these results, the most effective inhibitors of the sPLA2 enzymatic activity were MeOH (Figure 2A), Hex (Figure 2B), DCM (Figure 2C), and EtOAc (Figure 2D) extracts; *n*-BuOH (Figure 2E) and HE extracts (Figure 2F) did not show significant inhibition of sPLA2 enzymatic activity.

### 2.3. Prior HPLC Analysis of Methanol Extract with sPLA2

After screening the extracts for pharmacological and enzyme activities, those with higher inhibition of enzyme activity of *Crotalus durissus terrificus* sPLA2 were submitted to HPLC analysis. Methanol extract incubated with sPLA2 showed the greatest change in the profile before and after incubation. This extract indicated the presence of higher concentrations of components than the other extracts (Figure 3). Therefore, this extract was chosen for the isolation process for examination of bioaffinity with Cdt sPLA2.

### 2.4. Identifying Compounds from Extracts Through sPLA2 Protein–Ligand Approach

Compounds present in methanol extract were previously isolated from chromatographic procedures and identified using NMR analysis [12,13]. We tested the effects of unretained and retained compounds during the process of ultrafiltration from methanol extract incubated with sPLA2, as described in Section 4.4. From this procedure, the unretained fraction composed of compounds that did not interact with sPLA2 was washed and called MeOH 1C. The MeOH 2C fraction represents the compounds that bonded to sPLA2.

Figure 4 shows the overlap of chromatograms from sample MeOH 1C, MeOH 2C, and MeOH extracts. The MeOH 1C fraction showed compounds not retained by sPLA2, i.e., several caffeoylquinic acid derivatives with predominance of mono- and dicaffeoylquinic acids. The MeOH 2C fraction exhibited the presence of compounds retained by sPLA2. Analyzing the chemical composition of this fraction, the flavonoid hispidulin was identified as the major component (Figure 4, fraction MeOH 2C).

Both fractions were tested on the enzymatic activity of sPLA2 from *Crotalus durissus terrificus*. Figure 5A shows the MeOH 1C fraction containing the unretained compounds with higher inhibitory capacity than the MeOH 2C fraction containing the retained hispidulin. Figure 5B,C depicts both fractions (MeOH 1C and MeOH 2C) when injected 10 min after the inoculation of sPLA2 of *Crotalus durissus terrificus*. Figure 5B shows that the MeOH 2C fraction considerably abolished the pro-inflammatory effects of sPLA2 and the MeOH 1C fraction also significantly decreased the pro-inflammatory activity of sPLA2 but less effectively than the MeOH 2C fraction. Figure 5C indicates the MeOH 2C fraction two-fold decreased the myotoxic activity induced by sPLA2 in relation to the MeOH 1C fraction.

### 2.5. Characterization of Structure and Function of Hispidulin

Hispidulin (6-methoxy-5,7,4′-trihydroxyflavone) (His) is a polyphenolic compound belonging to the flavonoids. The presence of methylation at the C6 position jointly with a hydroxyl group at C7, both found in the hispidulin structure, is one of the most essential functional groups involved in various biological activities [19]. Thus, to investigate the importance of these two groups in modulating the pharmacological and biological activities of sPLA2, we compared the action of hispidulin with cirsimaritin (Cis) and scutellarein (Scu) under the same test conditions.

Cis has two methoxy groups in its structure, one of them in the same position as hispidulin (Figure 6A). Scu has no methoxyl groups in the structure, with hydroxyl groups at C6 and C7 positions. The hydroxyl groups at C6 and C7 positions in the flavonoid structure [20], as found in Scu, are the groups involved in the strong interaction with the *Naja* spp. snake venom sPLA2 and in the inhibition of the inflammatory activity induced by carrageenan. The presence of methoxyl groups in both Cis and His was not able to directly inhibit the enzymatic activity of sPLA2, unlike Scu, which almost completely abolished the enzyme activity of sPLA2 (Figure 6B). However, Scu decreased the peak of sPLA2-induced edema but was not able to effectively decrease the course of edema (Figure 6C). The methoxyl group in the C6 position of His and in the C6 and C7 positions of Cis is crucial for inhibiting the edematogenic activity induced by sPLA2. The results in Figure 6C suggest the methoxyl group in the C6 position could be crucial for neutralizing the pro-inflammatory activity of sPLA2 and this group may be responsible for sPLA2-induced inhibition of myotoxic activity (Figure 6D).

### 2.6. Molecular Interaction Analysis between sPLA2 and His

Figure 7 depicts how hispidulin is able to interact with sPLA2. Figure 7A shows the incubation product of sPLA2 with hispidulin when submitted to the same chromatographic conditions used to analyze the His profiles isolated in this study, sPLA2, and sPLA2 incubated with His. Figure 7A shows that the compound interaction with sPLA2 can be undone in the presence of polar aprotic solvents, such as acetonitrile used in this study. The results in Figure 7B suggest that His, when incubated with sPLA2, induces significant changes in the spectrum of sPLA2 circular dichroism (CD), mainly affecting the relative percentage of secondary structures in alpha-helices. The compound concentration used in sPLA2 incubation did not show any evident spectrum on CD. Figure 7C provides the results of molecular modeling of sPLA2 with His. The flavonoid is able to interact in the crack of the molecular region of Cdt sPLA2 among helices A, B, and D, which is a region located on the hydrophilic surface of sPLA2.

## 3. Discussion

The literature shows that methylation of hydroxyl groups from flavonoids can increase the structure’s potential as a pharmaceutical agent, leading to new applications. Flavonoid methylation can increase their metabolic stability and improve membrane transport, leading to better absorption and highly increased oral bioavailability [21]. Hispidulin is a flavone found in plants and is a promising candidate for the development of herbal medicine due to its antioxidant, antifungal, anti-inflammatory, anti-mutagenic, and antineoplastic properties [22]. Here, hispidulin showed promising activity against acute inflammation induced by sPLA2 and its identification was possible using a new method bioguided by molecular and functional bioaffinity against sPLA2 from *Crotalus durissus terrificus* to identify some potential secondary plant metabolites in various extracts. Thus, results showed that bioaffinity fishery on sPLA2 can provide a new alternative method for screening and capturing potential secondary metabolites with anti-inflammatory activity, as well as a tool for targeting the purification of compounds on a large scale.

Our results showed the method that was used to isolate hispidulin from methanol extract, which satisfactorily met some prerequisites such as decreased enzymatic activity, chromatographic changes in the extract with and without sPLA2, and decreased pharmacological activity with emphasis on edema. Although this process seeks secondary metabolites against edema, phospholipid catabolism is the limiting stage of the rate in AA generation, and, by the action of COXs and LOX, generates eicosanoids, which, in addition to participating in several cellular physiological processes, are involved in pathological processes such as inflammation [9,23]. Thus, the logic that we adopted in the development of this process was to analyze the action of extracts on enzymatic activity, which is cheaper than starting screening for pharmacological activity. Our data reinforce the idea that the development of new protocols for purification and isolation of natural compounds using sPLA2 as molecular bait requires a previous screening and that the investigative screening initiated by the edematogenic activity can be discarded for future isolation tests. The chromatographic data showed coherence between the chromatographic profile of the MeOH 1C and MeOH 2C fractions with the activity of these compounds on the edema and myotoxicity induced by sPLA2. Hispidulin was crucial for decreasing edema and myonecrosis.

The results presented in Figure 6 first demonstrated the effect of hispidulin isolated from the MeOH 2C fractions, and Figure 6A–D showed that hispidulin inhibited the edematogenic and myotoxic activity of sPLA2 and this inhibition does not involve a decrease in sPLA2. The presence of C6 methylation in ring A, which is crucial for reducing edematogenic and myotoxic activities, and the replacement of C6 methylation in ring A by a hydroxyl group, which considerably increases the inhibition of enzyme activity of sPLA2, did not significantly decrease the edematogenic activity and effectively reduced the myotoxic activity of sPLA2. C7 methylation of ring A did not improve hispidulin ability to inhibit enzyme as well as pharmacological activities induced by sPLA2. The results presented in Figure 5 and Figure 6 suggested that hispidulin did not inhibit the enzymatic activity of sPLA2, probably due to the interaction of hispidulin in a molecular region of sPLA2 including propeller A, propeller B, and helix D (Figure 7C and Figure 8). The compound does not interact with the active site of sPLA2, which is predominantly hydrophobic (Figure 8) and the interaction of sPLA2 with hispidulin is labeled and can be disrupted by aprotic polar solvents, which are used to solubilize polar compounds. The solvent used easily reverse the molecular interaction of sPLA2 with hispidulin (Figure 7A). The hydroxyl groups at C7 and/or the oxygen atom in methoxyl groups could be responsible for the hydrogen bridge with polar amino acids showing acidic or basic properties [24]. These characteristics are compatible with those observed in Figure 7C and Figure 8.

Figure 7B, the results of circular dichroism, shows that His induced a significant decrease in the relative percentage of its α-helix when incubated with sPLA2. This may be due to hydrogen bridges in the region of sPLA2 interaction (Figure 8). They could transiently change the secondary structure of sPLA2 and do not affect the enzymatic activity of sPLA2. Thus, the first conclusion is that the structural changes would not be permanent; these changes were significant for the loss of the enzyme activity of sPLA2 and influence the ability of sPLA2 to interact with the receptor. Several studies showed that the region of the calcium-binding loop close to the catalytic region of sPLA2 is known as the region involved in the interaction of sPLA2 with its receptor.

Our results also showed that hispidulin exerts its anti-inflammatory activity through its intracellular antioxidant action due to the presence of methylation. As such, His has the ability to pass through the plasma membrane and increase the antioxidant defense of the cell [8]. Numerous articles suggest that His exhibits a moderate antioxidant activity for free radical scavenging and neutralization, determined by 1,1-Diphenyl-2-picrylhydrazyl radical, 2,2-Diphenyl-1-(2,4,6-trinitrophenyl)hydrazyl (DPPH), and 2,2′-Azino-bis(3-ethylbenzothiazoline-6-sulfonic acid) diammonium salt (ABTS) antioxidant activity monitoring assays [12]. The presence of a carbonyl functional group at the C4 position plays an essential role in the neutralizing capacity of the hydroxyl radical [25]. Hydroxyl radicals are the most reactive of the free radical molecules, harming cell membranes and lipoproteins via lipid peroxidation, and are potential inducers of damage to DNA and other organic molecules such as proteins. Hydroxyl radicals can reduce disulfide bonds in proteins [26]. The results for hispidulin suggest that inflammation or myotoxic activity induced by sPLA2 may involve the formation of hydroxyl radicals as well as free radicals [8,9]. Thus, we were able to identify compounds that inhibit sPLA2 using the bioaffinity technique.

## 4. Materials and Methods

### 4.1. Materials

*Crotalus durissus terrificus* whole dried venom was purchased from Bio-Agents Serpentary (Batatais, São Paulo, Brazil). The solvents, chemicals, and reagents used for protein purification and characterization by HPLC were acquired from Sigma-Aldrich Chemicals (St. Louis, MO, USA), Merck (Kenilworth, NJ, USA), and Bio-Rad (Hercules, CA, USA). Male Swiss mice (25 g) were obtained from the Multidisciplinary Center for Biological Research (CEMIB) of the State University of Campinas (UNICAMP, São Paulo, Brazil). The animals were maintained under standard conditions (22 ± 2 °C; 12 h light/dark cycle) with food and water available ad libitum. All animal experiments were performed in accordance with Brazilian Laws for the Care and Use of Laboratory Animals, and the protocols were approved under Protocol No. 014-CEUA (23 August 2016) and Protocol No. 019-CEUA (23 August 2016).

Analytical HPLC analyses were performed on an Agilent 1260 chromatograph (5301 Stevens Creek Blvd., Santa Clara, CA, USA) equipped with an ultraviolet spectrum scanning detector by arrangement of photodiodes with a 60 mm flow cell. Reverse-phase C18 (Zorbax, Agilent, 4.6 × 150 mm, 5301 Stevens Creek Blvd., Santa Clara, CA, USA) with a 3.5 μm particle diameter was used as the stationary phase and a flow rate of 1.0 mL min^−1^ was employed for analysis on analytical scale. For separation of compounds, an Agilent HPLC 1200 semi-preparative chromatograph system was used with a C_18_ Zorbax Eclipse plus LC-18 column (25 cm × 10 mm) with 5 μm diameter particles and a flow rate of 4.18 mL min^−1^. HPLC grade solvents from T.J. Baker-Fisher Scientific (Bishop Meadow Road, Loughborough, Leicestershire, LE11 5RG) were used for the HPLC analyses. ^1^H and ^13^C-NMR spectra were recorded at 300 and 75 MHz, respectively, on a Bruker (Billerica, MA, USA) DPX-300 spectrometer. HMBCs were recorded on a Bruker Avance DRX-500 spectrometer (Bruker Corporation, Billerica, MA, USA). CD_3_OD or DMSO-d_6_ containing tetramethylsilane (TMS) as internal standard were used as solvents. Chemical shifts are reported in δ (ppm) and coupling constants (J) in Hz.

### 4.2. Purification of Secretory Phospholipase A2

Fractionation of phospholipase A2 from the total venom of *Crotalus durissus terrificus* was purified by two steps. In the first step, approximately 20 mg total venom was dissolved in 400 μL of 0.2 M bicarbonate ammonium buffer at pH 7.8 and shaken and centrifuged at 4500 × *g* for 5 min, and the supernatant was then injected into a molecular exclusion HPLC column (Superdex 75, 1 × 60 cm, Pharmacia, London, UK). The molecular exclusion column was also previously balanced with approximately 2 volumes of 0.2 M bicarbonate ammonium buffer (pH 7.8) and the chromatographic run was performed with an isocratic flow of 0.2 mL/min for the elution of the fractions that were collected at intervals of 400 μL per tube, and samples of 10 μL were set aside to monitor A2 phospholipase activity and separation of the crotoxin fraction. After confirmation of phospholipase enzymatic activity A2, the purified crotoxin fraction in the molecular exclusion column was collected in pools, lyophilized, and stored for the next chromatographic step. For sPLA2 purification, we used a C5 reverse-phase column (Jupiter 5 µm C5 300 Å, LC Column 250 × 4.6 mm, Phenomenex, 411 Madrid Avenue, Torrance, CA, USA). The column was previously balanced with buffer A (0.1% TFA (trifluoroacetic acid), pH 2.5) for 10 min and 200 µL samples were used. The chromatographic run was performed using a linear gradient of buffer B (66% acetonitrile prepared using a 0.1% TFA solution) and monitored using a wavelength of 280 nm, and the gradient used for the fractionation was buffer A for 5 min followed by a linear gradient of buffer B (45 min). The fractions of sPLA2 were then submitted to enzymatic assays and analyzed in electrophoresis in sodium dodecyl sulfate-polyacrylamide gel electrophoresis (SDS-PAGE). For sPLA2 incubation procedures, the extracts or compounds were applied in 10 µL samples, which were prepared using pure sPLA2 (1 mg/mL) with 10 µL of extract (0.1 mg/mL) at a ratio of 1:1 (*v*:*v*) and incubated for 30 min at room temperature.

### 4.3. Plant Material, Extraction Characterization of Compounds of Aerial Parts of Moquiniastrum Gloribundum

*M. floribundum* (Cabrera) G. Sancho was collected from Licínio de Almeida, Bahia, Brazil in January 2015. The plant was identified by Dr. Nádia Roque, and a voucher specimen which is deposited at ALCB Herbarium of Biology Institute of Federal University of Bahia (Bahia, Brazil). The plant was selected because some species of *Moquiniastrum* are used in traditional medicine, such as *M. polymorphum*. However, other species are poorly studied in relation to their bioactivities. Therefore, leaves of *M. floribundum* were dried, powdered, and extracted with *n*-hexane and subsequently with methanol. MeOH was resuspended in MeOH:H_2_O (7:3) and partitioned successively with *n*-hexane, DCM, EtOAc, and BuOH. The material not extracted with those solvents formed the hydroalcoholic (HA) phase.

A semipreparative HPLC separation was conducted with the components from the MeOH extract. The following elution gradient was employed with a flow rate of 4.18 mL min^−1^: solvent A (Type 1 ultrapure water acidified with 0.1% acetic acid (*v*/*v*)) and solvent B (acetonitrile (ACN)), and the ratios were 0–3 min: 10%–20% B; 3–7 min, 20% B; 7–8 min, 20%–25% B; 8–12 min, 25% B; 12–17 min 25%–50% B; 17–22 min, 50%–100% B; 22–45 min, 100% B. The column temperature was 45 °C, the injection volume of sample was 200 μL, and the sample was dissolved in methanol at concentration of 100 g L^−1^. Using this procedure, we were able to isolate, through one chromatographic run, the compounds presented in the extract. These compounds were identified using NMR (^1^H, ^13^C, and HMBC spectra) as 3-caffeoylquinic acid (**1**), chlorogenic acid (**2**), 4-caffeoylquinic acid (**3**), 3,4-, 3,5-m, and 4,5-dicaffeoylquinic acids (**4**–**6**, respectively), 3,4,5-tricaffeoylquinic acid (**7**), and hispidulin (**8**). The proposed structures were confirmed by the comparison of the spectroscopic data with those reported in the literature [12,13]. Subsequent component identification was performed using the retention time of each compound.

### 4.4. Bioaffinity Fishing of Natural Products

HPLC-MS coupled with bioaffinity ultrafiltration for screening natural compounds was completed in three basic steps: (1) incubation of the target protein with a mixture of natural compounds and the establishment of the interaction of the compounds with the target protein, (2) ultrafiltration of the compounds bound to the target protein, and (3) compound identification.

Despite being an easy and reproducible technique, this method involves some technical and limiting difficulties. One of the difficulties is related to the high protein concentration required to capture the different compounds present in the sample or botanical extract, which is incubated with the target protein for 15 min to 2 h at 25 °C. The second limiting point is the choice of buffer, which may vary from phosphate buffer, Tris-HCl, ammonium acetate, and addition of organic solvents (generally 10% isopropanol) to decrease a large number of nonspecific interactions. Following the incubation, ultrafiltration is performed using a regenerated cellulose membrane with a protein retention porosity of 10,000 Da molecular weight, allowing separation of unbonded compounds and bonded proteins. Bonded compounds are either analyzed directly by mass spectrometry or retained on an HPLC column and analyzed by HPLC-MS.

The literature does not report any protocol to identify and capture secondary metabolites using snake venom sPLA2. Therefore, we used isolated sPLA2 purified from Cdt venom for protein–ligand fishing from extracts of natural products. The procedure described here was developed as an alternative tool for screening of potential compounds with affinity to sPLA2 from Cdt. We also identified other potential secondary metabolites that may modulate the enzymatic and mainly pro-inflammatory activity induced by sPLA2. According to Figure 9, this new process developed using the enzymatic and pharmacological properties of sPLA2 of Cdt can be modified and scaled for specific applications.

Essentially, the extracts are prepared, lyophilized, and subsequently dissolved in an aqueous saline solution, such as a solution of ammonium bicarbonate (AMBIC), and buffer was also used for the preparation of sPLA2 solution of *Crotalus durissus terrificus* (Cdt). The extract solution was dissolved in the aqueous solution, vigorously mixed, and centrifuged at 4500 × *g,* and the supernatant was collected and mixed with the sPLA2 solution. Thereafter, this sPLA2 mixture plus extracts were incubated for 30 min at room temperature. After this incubation period, the mixture was placed in an ultrafiltration apparatus using a 10 kDa molecular mass cut off concentration membranes. This mixture was first centrifuged to elute the samples not bound to sPLA2, which was named 1C. Then, the retained material was dissolved and washed again with AMBIC solution. Subsequently, the retained material was dissolved with a new washing solution (AMBIC, acetonitrile 15%, and Dimethyl sulfoxide (DMSO) 5%), followed by a second wash to elute the retained material, which allowed elution of the 2C fraction by centrifugation. After obtaining the fractions, they were lyophilized and stored for later use.

### 4.5. Enzymatic Assays

Phospholipase A2 and total venom activity were initially quantified using the chromogenic substrate 4N3OBA (NOBA), which was dissolved in acetonitrile (100%). The venom and purified sPLA2 had the concentration adjusted to 1 mg/mL and was added to 175 µL of buffer. The samples were incubated in 96-well plates at room temperature. Thirty minutes later, the absorbance at 425 and 600 nm (to correct for any turbidity in the sample) were determined on a microplate reader. Once the enzymatic activity was verified, the ultrafiltration of compounds was completed and the sample was evaluated for sPLA2 inhibitory activity. After the enzymatic assays, a simple reverse-phase HPLC analysis was done using an analytical Phenomenex C5 column (Phenomenex, International, 411 Madrid Avenue, Torrance, CA 90501-1430, USA). To analyze the interaction of the extracts with the protein, each extract was incubated with sPLA2 and posteriorly assayed on HPLC. For 30 min at room temperature, 10 μL of pure sPLA2 (1 mg/mL) was incubated with 10 μL of each extract, methanol (MeOH + sPLA2), hexane (Hex + sPLA2), and dichloromethane (DCM + sPLA2) at concentration of 0.1 mg/mL and ratio of 1:1 (*v*:*v*).

### 4.6. Circular Dichroism (CD)

A J-815 CD Spectrometer (Jasco, São Paulo, Brazil) was used to evaluate the secondary structure of proteins (α helices, β sheets, turns, and random secondary structures) in solution. This is a fast and economical method due to the small quantities of samples required. In this technique, polarized light is used in the distal ultraviolet (UV) range from 180 to 260 nm. This technique allows the evaluation of the structural integrity of proteins, conformational changes, and processes of denaturation (unfolding) and renaturation (folding), allowing the estimation of the composition of the elements in the secondary structure of this macromolecule. The CD analysis was performed with 0.1 mg/mL of sPLA2, and the compounds prepared according to Section 2.2. Isolated compounds were also used to evaluate possible interference in the tests. The analyses were performed to eight convolutions, and data were treated by Spectra Manager software (Jasco, 28600 Mary’s Court Easton, MD, USA).

### 4.7. Pharmacological Assay and Biochemical Assays

#### 4.7.1. Paw Edema

A paw edema assay was performed using male Swiss mice (25 g). Posterior paw edema was induced by a single subplantar injection of sPLA2. Paw volumes were measured immediately before the injection of the samples and at selected time intervals thereafter (0, 30, 60, 180, and 240 min) using a hydroplethysmometer (model 7150, Ugo Basile, Monvalle, Italy). The results are expressed as the increase in paw volume (mL) calculated by subtracting the initial volume. Each treatment was conducted for *n* = 5. After the tests, the mice were anesthetized and sacrificed via cervical dislocation. In this protocol, sPLA2 was incubated in presence of extract at volume ration of 1:1 (*v*:*v*), where sPLA2 concentration was adjusted to 1 mg/ mL, the extract was adjusted to 0.1 mg/ mL, and both were incubated together for 30 min at room temperature.

#### 4.7.2. Evaluation of Myonecrosis

The myotoxic activity was evaluated using the plasma creatine kinase (CK) measurement released from damaged muscle cells, using a commercial CK-NAc kit (Laborlab, London, UK). Briefly, the right gastrocnemius muscle was injected with 50 μL of 0.5 mg/mL sPLA2 sample, whereas the control mice received an equal volume of 0.15 M NaCl. After 3 h, the animals were anesthetized and samples were collected from the abdominal cavity into tubes containing heparin as an anticoagulant. The plasma was stored at −10 °C for a maximum of 12 h before the assay. The level of CK was determined with 40 μL of plasma, which was incubated for 3 min at 37 °C with 1.0 mL of the reagent according to the protocol kit. The resulting activity is expressed in U/L. Each treatment was conducted for *n* = 5. After the tests, the mice were anesthetized and sacrificed via cervical dislocation.

### 4.8. Structural Modeling of the Hispidulin–sPLA2 Complex

Structural modeling of the hispidulin binding to the Cdt sPLA2 active site was completed using the crystallographic structure determined by Marchi-Salvador et al. [27] (PDB: 2QOG). First, the Cdt sPLA2 enzyme structure was aligned with sPLA2 isoforms from different snake species deposited in the Protein Databank (PDB: 2QOG). Only enzymes possessing high structural identity (C α root mean square deviations < 2.0 Å) and presenting ligands/inhibitors in the active site were used to modeling approaches. The sPLA2 from the following species matched these criteria: *Agkistrodon halys Pallas* (PDB code: 1B4W), *Bothrops jararacussu*, (1Z76), *Bothrops moojen* (1XXS), *Daboia russelli pulchella* (1OYF, 4QER, 4QGD, 4QF8, 4QF7, 4QEM), and *Vipera nikolskii* (2I0U). The protein structural alignments were performed using the Gesamt tool in the CCP4 package [28] and the hispidulin molecular modeling and normalization was performed using COOT [29] based in the positions of similar chemical groups of the ligands/inhibitors present in the sPLA2 active sites structures (β-octylglucoside, *p*-bromophenacyl bromide, 6-methyl heptanol, resveratrol, gramine, spermidine, corticosterone, *p*-coumaric acid, 2-(acetyloxy) benzoic acid, octyl phenol ethoxylate, and stearic acid). Coot is a molecular-graphics application for model building and validation of biological macromolecules [29].

### 4.9. Statistical Analysis

Results are reported as the means ± SEM of replicated experiments. The significance of differences between means was assessed by an analysis of variance (ANOVA) followed by Dunnett’s test when several experimental groups were compared to the control group. The confidence limit for significance was 5%.

## Figures and Tables

**Figure 1 molecules-25-00282-f001:**
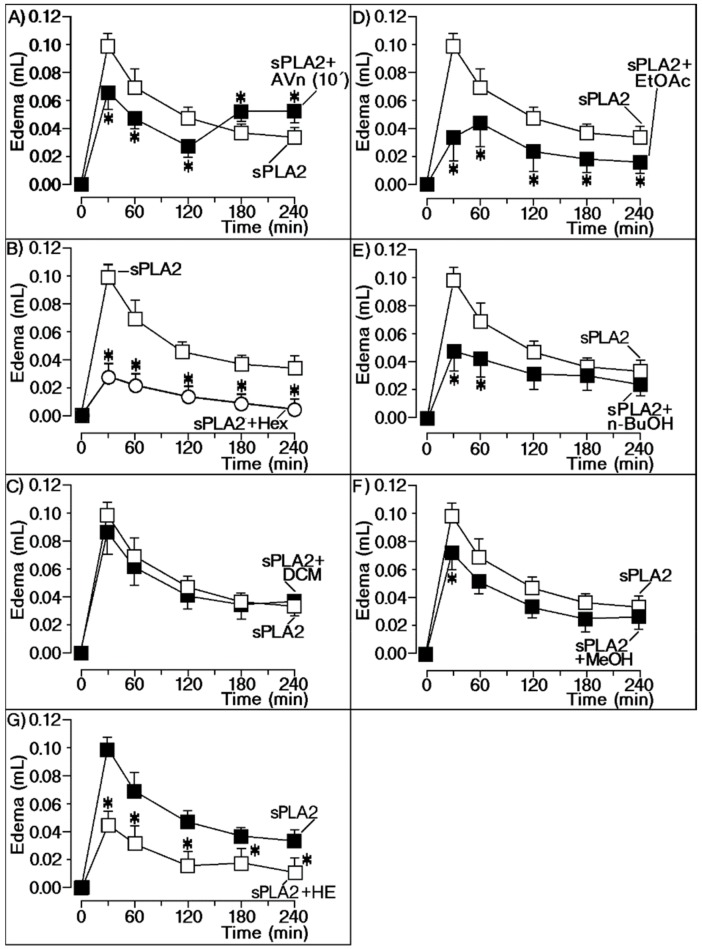
Extracts screening against edematogenic activity of native secretory phospholipase A2 (sPLA2). All panels in this figure show the evaluation of extracts against sPLA2-induced edema with seven different extracts. (**A**) Effect of injected antivenom (AVn) 10 min after application of Cdt sPLA2, which significantly decreased the pro-inflammatory action induced by sPLA2; (**B**) hexanic (Hex), (**D**) ethyl acetate (AcOEt), (**E**) *n*-butanolic (*n*-BuOH), and (**G**) hydroethanolic (HE) extracts significantly reduced sPLA2-induced edema. The results strongly suggested that these extracts have a potential anti-inflammatory compound interacting with sPLA2. (**C**) Dichloromethane (DCM) and (**F**) methanol (MeOH) extracts demonstrated no significant effects on sPLA2 edematogenic activity. * significant differences relative to a standard. All analyses were performed using analysis of variance (ANOVA, *p* < 0.05), and each bar represent *n* = 5.

**Figure 2 molecules-25-00282-f002:**
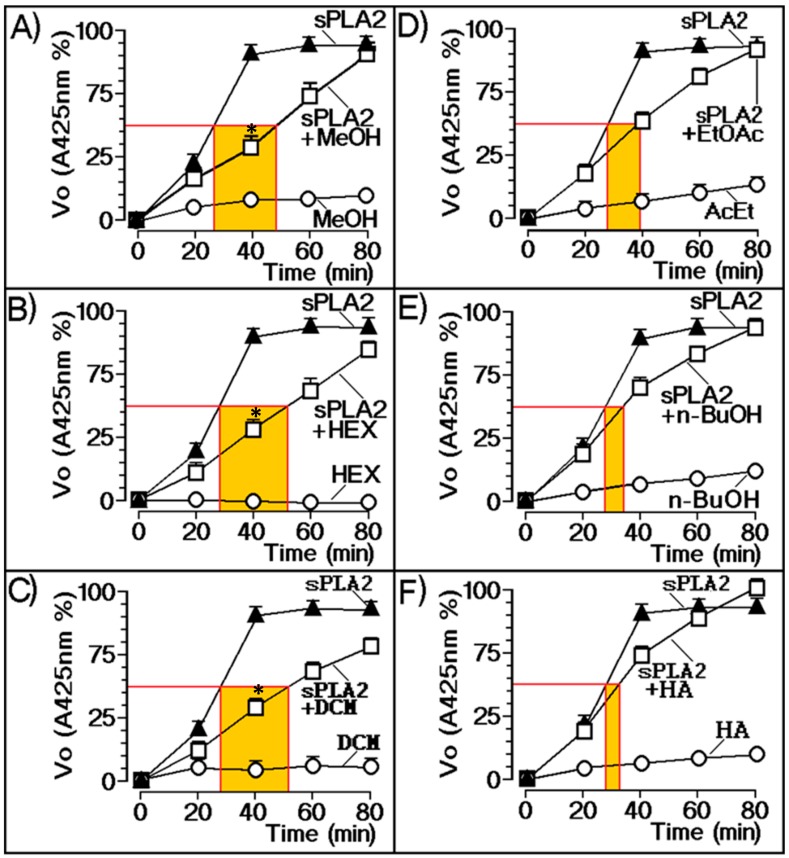
Extract evaluation against enzymatic activity of native sPLA2. (**A**) Methanol (MeOH), (**B**) hexane (HEX), (**C**) dichloromethane (DCM), and (**D**) ethyl acetate (EtOAc) extracts exhibited inhibition of enzymatic activity. (**E**) *n*-butanol (*n*-BuOH) and (**F**) hydroethanolic extracts (HA) showed no significant inhibition of sPLA2 enzymatic activity. Error bars indicate the standard error of the mean (SEM); ∗ *p* ≤ 0.05 compared with the saline control.

**Figure 3 molecules-25-00282-f003:**
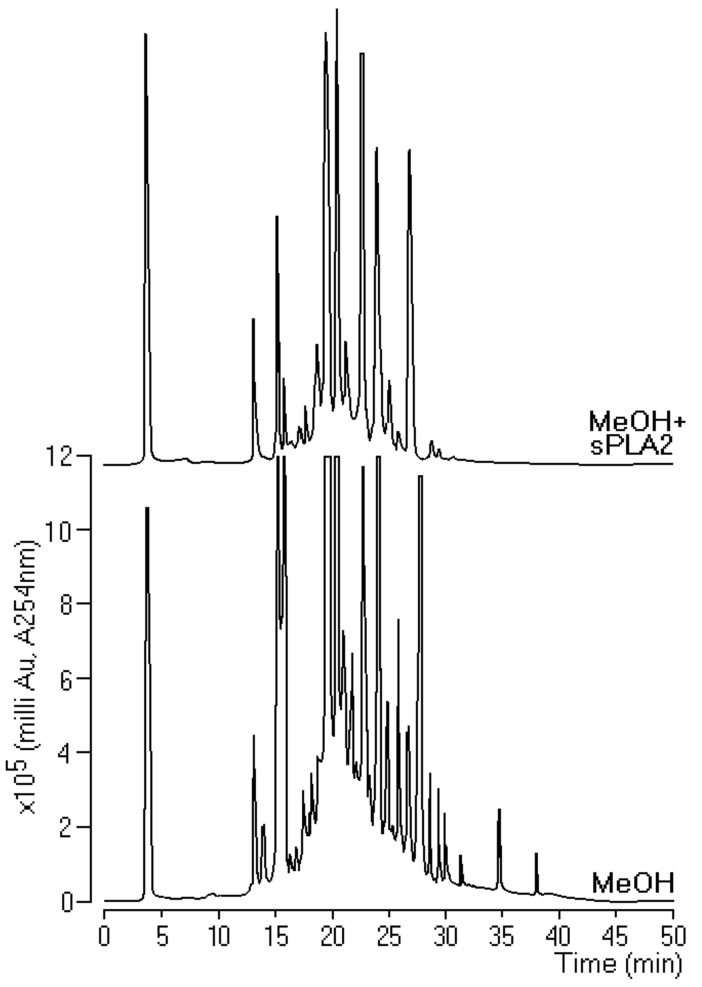
Reverse-phase HPLC analysis using an analytical Phenomenex C5 column where we incubated 10 μL of pure sPLA2 (1 mg/mL) with 10 μL of extract (0.1 mg/mL) at a ratio of 1:1 (*v*:*v*).

**Figure 4 molecules-25-00282-f004:**
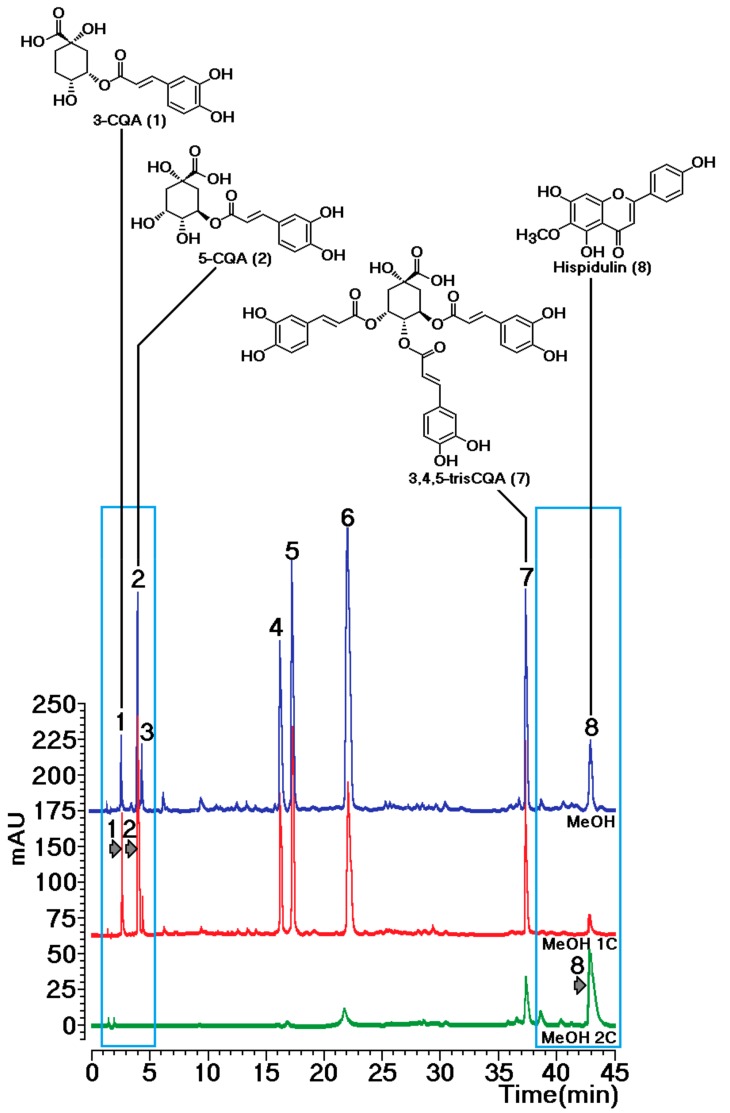
Overlap of chromatograms from samples of methanol crude extract and MeOH 1C and MeOH 2C fractions obtained through the sPLA2 protein–ligand approach. Compounds were identified through NMR analysis as previously described [12,13]: (1) 3-CQA, 3-caffeoylquinic acid; (2) 5-CQA, 5-caffeoylquinic acid; (3) 4-CQA, 4-caffeoylquinic acid; (4) 3,4-dicaffeoylquinic acid; (5) 3,5-dicaffeoylquinic acid; (6) 4,5-dicaffeoylquinic acid; (7) 3,4,5-triCQA, 3,4,5-tricaffeoylquinic acid; and (8) hispidulin.

**Figure 5 molecules-25-00282-f005:**
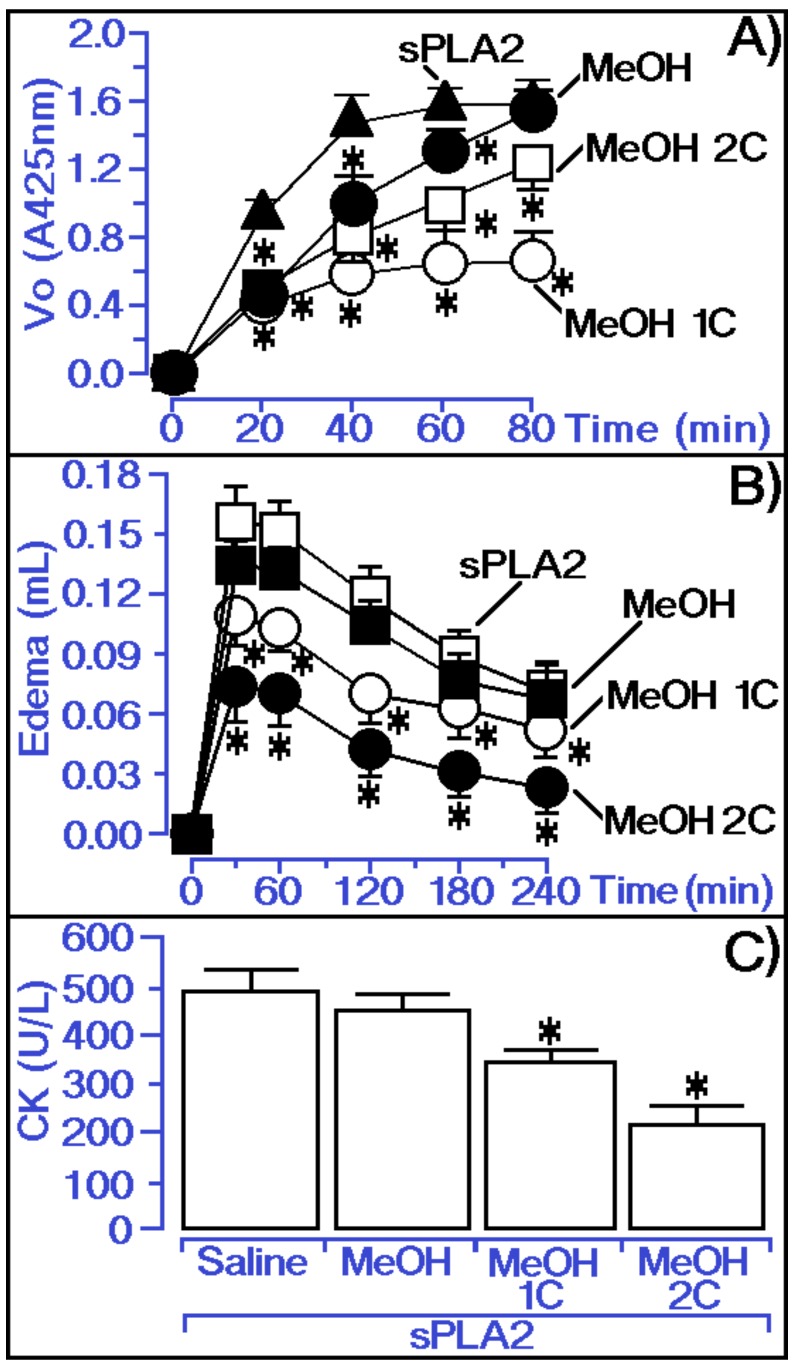
Comparative analysis of 1C and 2C fractions from methanol extract incubated with sPLA2 on enzymatic (Figure 5A), edematogenic (Figure 5B), and myotoxic activities (Figure 5C) of isolated sPLA2. (**A**) Activity of the MeOH, MeOH 1C and 2C on phospholipase A2 enzymatic activity, where saline represents the control group. The results are expressed as a variation of enzymatic velocity Vo, and the error bars indicate the SEM. * Statistically significant differences (*n* = 5, *p* < 0.05) compared to native sPLA2. (**B**) show paw edema induced after the injection of sPLA2 and sPLA2 treated with MeOH extract and MeOH 1C and MeOH 2C, respectively, into the right paws of Swiss mice. Edema is expressed as volume in µL and measurements were performed after 30, 60, 120, 180, and 240 min. The error bars indicate the SEM of five experiments. * *p* < 0.05 compared to native sPLA2. (**C**) show the myonecrosis assay and the results are expressed as creatine kinase (CK) units of enzymatic activity per liter (U/L). The error bars indicate the SEM of five experiments. * *p* < 0.05 compared with native sPLA2.

**Figure 6 molecules-25-00282-f006:**
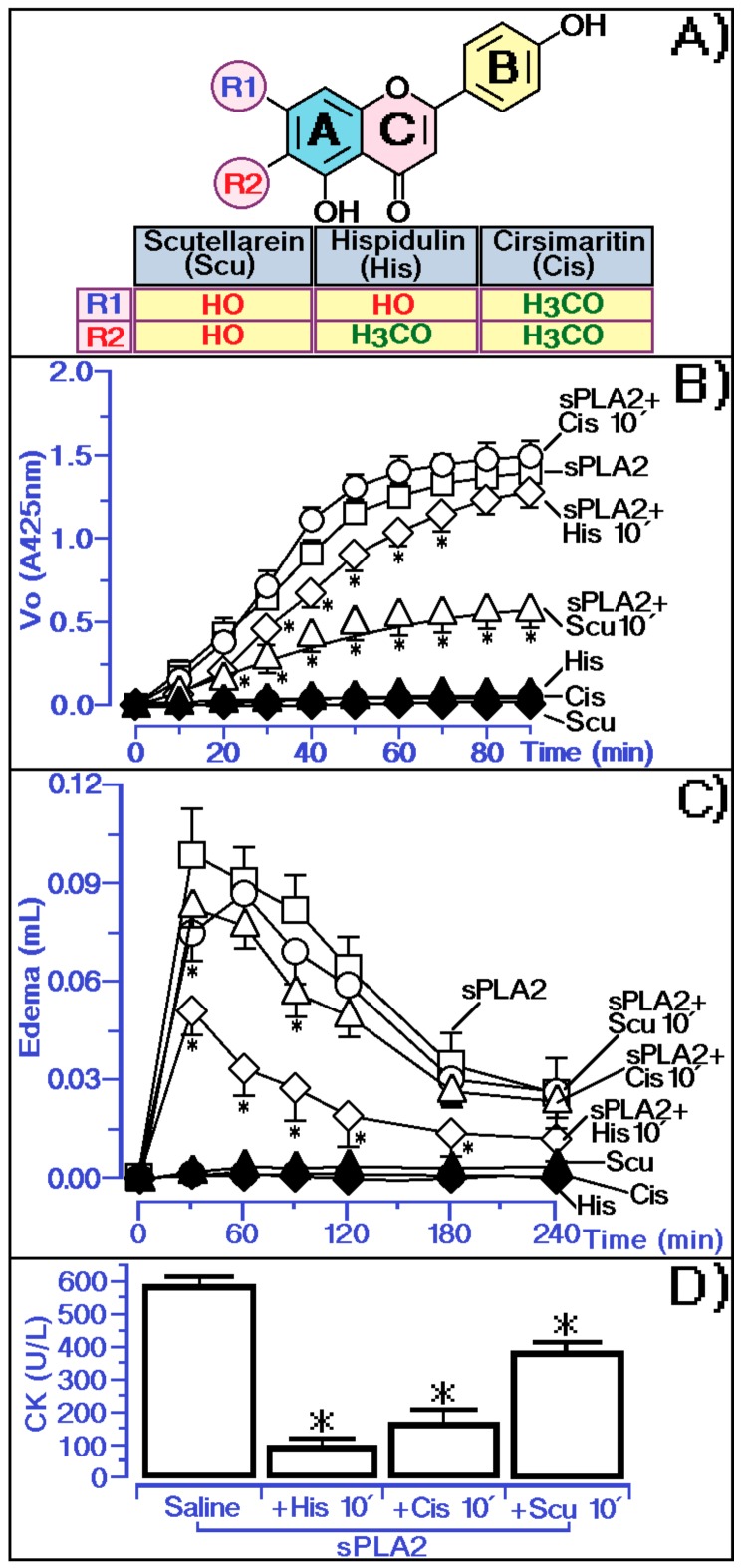
Analysis of the methyl and hydroxyl group on the protective function of hispidulin in inhibiting sPLA2-induced edema and myonecrosis and influences of these groups on the inhibition of sPLA2 enzymatic activity. (**A**) showed the structure of Hispidulin compared to Scutellarein (Scu) and Cirsimaritin (Cis). (**B**) Comparative analysis of His, Scu and Cis on phospholipase A2 enzymatic activity, where saline represents the control group. The results are expressed as a variation of enzymatic velocity Vo. (**C**) show paw edema induced after the injection of sPLA2 and sPLA2 treated with His, Cis and Scu, respectively, into the right paws of Swiss mice. Edema is expressed as volume in µL and measurements were performed after 30, 60, 120, 180, and 240 min. (**D**) show the myonecrosis assay and the results are expressed as creatine kinase (CK) units of enzymatic activity per liter (U/L). Error bars indicate the SEM; * *p* ≤ 0.05 compared with the saline control.

**Figure 7 molecules-25-00282-f007:**
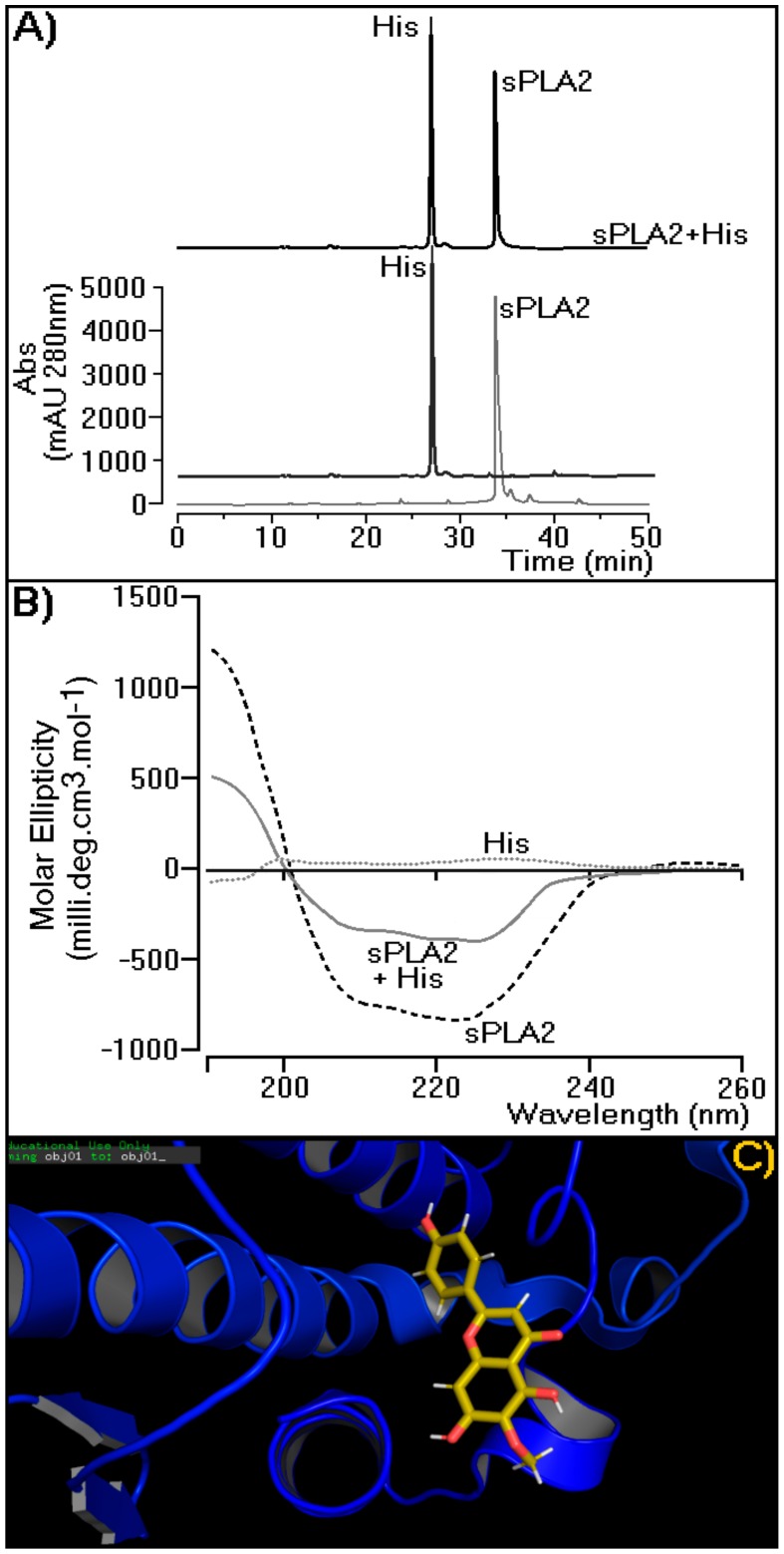
Characterization of possible molecular interaction of His with sPLA2 from *Crotalus durissus terrificus. (***A**) Investigation of the His–Ctd sPLA2 using reverse-phase HPLC. (**B**) Circular dichroism (CD) spectral profile data were acquired over the range of 185–280 nm. Ctd sPLA2 is represented by the gray line, Ctd sPLA2 + His by the dashed gray line, and His by the dotted gray line. The CD spectra are expressed in θ cm^2^/dmols. (**C**) Molecular modeling of hispidulin in the sPLA2 active site. The structure of Ctd SPLA2 is represented in dark blue (PDB: 2QOG) and the His molecule by sticks and lines with the atoms colored as follows: C, yellow; N, blue; O, red; and H, white (lines).

**Figure 8 molecules-25-00282-f008:**
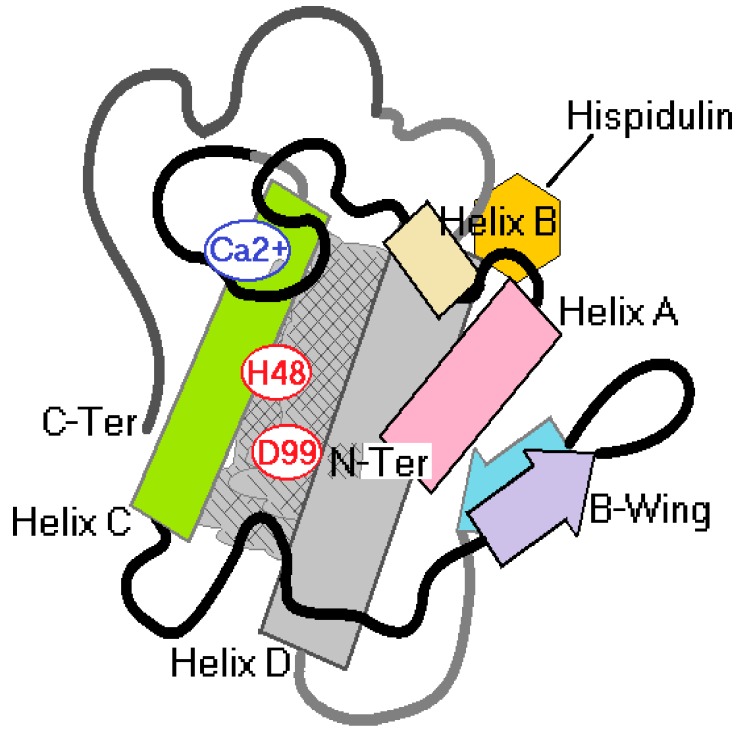
Schematic to explain a possible molecular interaction of hispidulin on sPLA2.

**Figure 9 molecules-25-00282-f009:**
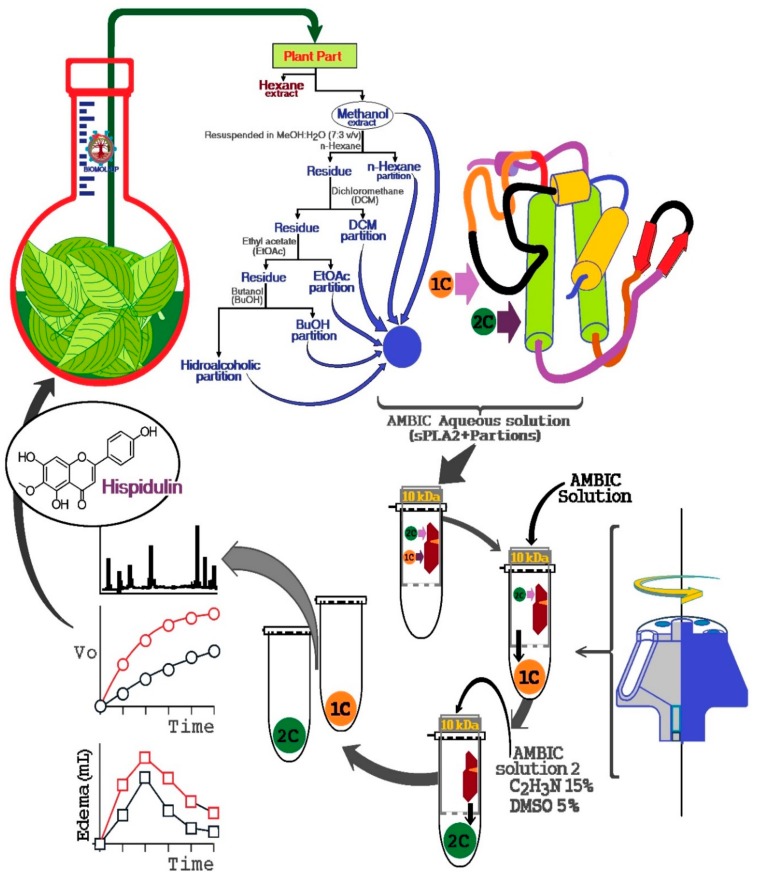
General scheme of screening natural compounds using secretory Phospholipase A2 as a target molecule for fishing for plant extract molecules and this method is intended to provide a tool to investigate potential molecules capable of modulating or inhibiting the pro-inflammatory actions induced by sPLA2 from plants or other organisms and to assess the importance of certain species among the biological diversity of a region as a potential producer of anti-inflammatory compounds.

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
