# Peer review of "Bioaffinity Fishing Procedure Using Secretory Phospholipase A2 for Screening for Bioactive Components: Modulation of Pharmacological Effect Induced by sPLA2 from Crotalus durissus terrificus by Hispidulin from Moquiniastrum floribundum"

_molecules, 2020, doi:10.3390/molecules25020282_

Round 1

Reviewer 1 Report

This study is giving a new insight and approach for the screening bioactive compounds from natural products. The way of research seems to be original and most of the experiment was performed logically, however, some part should be revised for the publication in this journal.

Major concern was related with identification of active compounds through the sPLA2 reaction. No information on the analysis and identification for the metabolites can be found, which need to be amended.

Figure legends should be revised for clear understanding the contents in figures.

Abbreviations should be indicated.

Fig. 2, 3: Abbreviations should be explained in the figure legends, i.e. DCM, EtOAc etc…

Line 145-150: Where are the data for the other extracts except for methanol extracts? How could author determine methanol extracts had higher concentrations of components? Did you chase the HPLC elution with just 254 nm, not for scanning? Authors should clarify the scientific basement to select methanol extracts for the following research. Metabolomic comparison also could be another way to go…

Line 175-176: How could you determine hispidulin as the major component?

Fig. 5: Similar inquiry… How did you determine these secondary metabolites? No description can be found in manuscript which should be indicated. In addition, abbreviations for the metabolites should be indicated in the figure legend.

Author Response

Manuscript ID: molecules-669319 - Revision

We have tried very hard to answer all the questions asked and we hope that the attached document may have cleared up most of the doubts.  After answering all the reviewers' answers, an extensive review of English will be carried out by contracting the services of MPDI.

Reviewer 2 Report

The article displays some notable results dealing with screening and method development studies using extracts of Moquiniastrum floribundum with anti-inflammatory potential. The idea is basically good, but the abstract and title needs to be revised for better representing the essence of manuscript; the description of results and methodology need to be improved.

While the manuscript is certainly of interest, several questions need to be addressed in order to prepare the manuscript for a possible publication.

Major points:

What were the selection criteria for the medicinal plant Moquiniastrum floribundum to be investigated? What do we know about the secondary metabolites other than flavonoids of Moquiniastrum floribundum? What could be the role of other compounds in the pharmacological activities observed? The extraction/separation method described is quite confused. There is a significant discrepancy between the figure summarizing the extraction and the text describing the method used. If someone looks at the figure he/she can observe that a methanol extraction was performed followed by solvent-solvent partition using hexane, dichloromethane, ethyl acetate, butanol and aqueous ethanol. In the text there is first an extraction with hexane, then an extraction with methanol. Which one is the right one? If the text is the correct one, it is not clear for me why the authors performed a hexane extraction before the methanol one? I do not really understand the reason for this.

According to the figure there was a partition between the (aqueous)methanol residue and aqueous ethanol, which is nonsense, because there are miscible with each other, they do not form two phases.

On the chromatogram of figure 5 we can see peaks 4, 5, and 6. What do we know about these compounds? Language of the manuscript requires improvement. There are several confusing paragraphs which should be clarified for better understanding

e.g.: Line 262,

Lines 269-276,

Lines 278-281

In the last paragraph the authors state that “hispidulin isolated in this study reveals several interesting qualities such as molecular stability, ability to neutralize hydroxyl radicals and that may be involved in the inflammatory… process”. These characteristics are valid for several flavonoids, and not only for hispidulin.

Some minor points:

instead of “Hispidulin” use “hispidulin” with small letter Figure 5: use “Phenomenex” instead of “phenomex” Use “butanol” instead of “buthanol” in figure 1. Please use “partition” instead of “partion” in figure 1. Line 133: “that both extracts did not have compounds” – needs to be revised Line 218: “The presence of methoxyl groups in both Cis and His were not able” should be “the presence … was not able” Figure 8: Crotalus durissus terrificus in italics Line 258: “hispidulin is a promising candidate” instead of “hispidulin has a promising candidate” Line 385: it is not necessary to have the flow rate with three decimals Line 396: Use “Bioaffinity” instead of “bioaffinity“ Line 407: acetonitrile instead of Acetonitrile

Author Response

(The authors gave the same response as above.)

Reviewer 3 Report

The manuscript reported an alternative screening method for the isolation of bioactive natural secondary metabolites through Bioaffinity fishing procedure. With the aim to discovery of new anti-inflammatory compounds, a secretory phospholipase A2 from total venom of Crotalus durissus terrificus was selected.

The following points should be addressed.

Abstract Lines 26-29. Reformulate the sentence since it is too descriptive for an abstract. Keywords Replace keywords hispidulin and sPLA2 since they are mentioned in the title. Introduction: Page 2 lines 47-52. The two sentences are redundant. Please reformulate. In figure 1 add 2C in the green circle. Moreover, it isn’t clear the meaning of the chemical structure reported. The procedure description of method in the introduction sounds unusual, move it as caption of Figure 1. Moreover, improve Figure 1 legend. Page 3 line 100. Add references Page 4 line 114. Use the abbreviations for phospholipase A2 and Crotalus durissus terrificus. Results: Page 4 line 126. Please, use inject or administration and not both. All figures all panels. Don’t label curves but use appropriate legend. Line 131-133: the panel 2F shows a significant value. Page 6 figure 3: improve figure legend. Add statistically significant Page 7 figure 4: improve figure legend. Page 7 line 185. What do authors mean with “virtually”? Page 9 figure 6: zoom out figure. Improve legend. Page 10 line 214. Delete the “presence of”. Page 10 line 219. Replace enzyme with enzymatic. Specify what authors mean with virtually. Page 10 line 225. Reformulate. Page 11 figure 7: improve figure legend. Discussion: Re-write discussion paragraph, because it seems a results analysis. Page13 line 259. Replace has with is. Page 13 line 262. “In this article we were able…” this beginning of sentence makes no sense. Please, rephrase. Page 13 line 267. Avoid repeating the word potential more times. Page 13 line 269. Avoid to use the following sentence “Briefly, the analysis of figures 2,3,4 and 5 shows..” since it sounds as a result comment. Materials and Methods: Page 16 line 379. Put the references at the end of the manuscript. Page 16 lines 381-383. Extraction procedure described is different from Figure 1 and the description in results. Page 16 line 388. 22-45 min (??) chromatographic course is 45 min. Page 18 line 443. Perhaps 240 min not 360 min. Page 18 lines 463 and 465. Use Cdt abbreviation. Page 18 lines 465, 472 and 473. Put the references at the end of the manuscript. Page 18 lines 467 and 475. Verify symbols

Author Response

(The authors gave the same response as above.)

Round 2

Reviewer 1 Report

Manuscript has been revised properly and clearly. Recommend to be published as it is.

Author Response

Author's response to review report (reviewer 1). Comments and suggestions for authors: The manuscript was reviewed appropriately and clearly. Recommend that it be published as is.  We greatly appreciate the help of the suggestions and we revised the text both in style and grammar revision and other suggestions from the academic reviewer as well as reviewer 2 were also contemplated and so in yellow are the major changes reviewed and follows the latest version of the article.   Attached is the revised article  

Reviewer 2 Report

Significantly improved, most of the required corrections made. English review planned.

Author Response

Dear Mr. Reviewer First of all, we thank you for the fair comments that have been made and that have helped us to put this last revision to your appreciation.  According to proofreader 2, in the comments and suggestions for Authors Significantly improved, most of the required corrections made. English review planned.  We improved the materials and methods part, reviewed all figures and made the correction of figure 1, which was fixed to follow in a coherent way with the other figures.  All the most significant changes are in yellow and MPDI's services were hired to help us revise the text in its form and point out points that deserved further consideration.  In the attached text that was revised, the corrections are in yellow.  The final text follows in the attachment.
